# Cost-Effective Copper–Nickel-Based Triboelectric Nanogenerator for Corrosion-Resistant and High-Output Self-Powered Wearable Electronic Systems

**DOI:** 10.3390/nano9050700

**Published:** 2019-05-05

**Authors:** Kequan Xia, Zhiwei Xu, Zhiyuan Zhu, Hongze Zhang, Yong Nie

**Affiliations:** 1Ocean College, Zhejiang University, Zhoushan 316021, China; 21734118@zju.edu.cn (K.X.); xuzw@zju.edu.cn (Z.X.); 0016180@zju.edu.cn (Z.Z.); 2Nanjing Electronic Devices Institute 524 East Zhongshan Road, Nanjing 210016, China; zhanghongze007@pku.edu.cn; 3School of Mechanical Engineering, Zhejiang University, Hangzhou 310027, China

**Keywords:** triboelectric nanogenerator, copper–nickel, corrosion-resistant, wearable electrics

## Abstract

Recent years, triboelectric nanogenerators (TENGs) have attracted increased attention from researchers worldwide. Owing to their conductivity and triboelectric characteristics, metal materials can be made as both triboelectric materials and conductive electrodes. However, the surface of typical metals (such as copper, aluminum, and iron) is likely to be corroded when the sweat generated by human-body movement drops on the surface of TENGs, as this corrosion is detrimental to the output performance of TENGs. In this work, we proposed a novel corrosion-resistant copper–nickel based TENG (CN-TENG). Copper–nickel alloy conductive tape and polytetrafluoroethylene (PTFE) tape played the role of the triboelectric materials, and polymethyl methacrylate (PMMA) was utilized as the supporting part. The conductive copper–nickel alloy tape also served as a conductive electrode. The open-circuit voltage (V_OC_) and short-circuit current (I_SC_) can arrive at 196.8 V and 6 μA, respectively. Furthermore, peak power density values of 45 μW/cm^2^ were realized for the CN-TENG. A series of experiments confirmed its corrosion-resistant property. The approximate value of V_OC_ for the fabricated TENG integrated into the shoe reached 1500 V, which is capable of driving at least 172 high-power LEDs in series. The results of this research provide a workable method for supporting corrosion-resistant self-powered wearable electronics.

## 1. Introduction

In recent years, wearable flexible smart electronics have attracted increasing attention due to the advantages, including low weight, convenience, and multi-functionality [1,2,3,4,5]. Moreover, the rapid development of micro/nanofabrication technology has promoted the ultra-miniaturization and super-integration of wearable electronics [6,7,8]. However, the power source of wearable electronics remains a bottleneck for their development. Traditional power sources (conventional chemical batteries) are typically large and lead to severe environmental pollution [9,10,11]. Further, with the continuous development of wireless internet and service upgrades, the energy storage capacity of the traditional batteries that have been used for portable electronics has become inadequate [12,13,14,15,16]. 

In previous work, some vibration energy harvesters based on piezoelectric and electromagnetic effects are used as considered promising power supply sources for micro-devices, such as the tunable multi-frequency vibration energy harvester [17], the structural damping and the electromechanical coupling [18], and the two-degree-of-freedom hybrid piezoelectric–electromagnetic energy harvester [19]. Electromagnetic vibration energy harvesters are widely used due to their small size and adaptability in various harsh environments. However, converting the human-body movement mechanical energy is difficult for previous energy harvesters [20]. Piezoelectric vibration energy harvesters suffer from broad environmental vibration bandwidth and high levels of randomness. Therefore, the harvester and conversion efficiency of these harvesters will decrease significantly [21,22,23]. 

In 2012, triboelectric nanogenerators (TENGs) were first proposed as an effective energy harvester of various mechanical motions, including mechanical vibration, human movement, raindrops, and air/water flow [24,25,26,27,28,29,30]. TENGs are new types of energy harvesters that are lightweight and flexible with high output performance [31,32,33,34,35]. They are also referred to as green batteries, as they generate less pollution than traditional batteries, such as lead–acid and Ni–Cd batteries [36,37]. Previous works have proposed TENGs that are based on vertical separation, which are defined as the dielectric-to-dielectric TENGs in vertical separation mode [38]. Subsequently, a new approach to prepare TENGs without the depositing metal electrodes are proposed and defined as the conductor-to-dielectric TENG in vertical separation mode [39].

However, the conductor-to-dielectric triboelectric mode requires electronic conductivity for one of a triboelectric pair. Thus, various metal materials including copper, aluminum, and iron [40,41,42] have been used for the fabrication of the conductor-to-dielectric TENG. It is obvious that the characteristics of the triboelectric materials’ surface are crucial for the electrical output of TENG, especially considering that the surface of metal materials becomes corroded when the sweat generated by human-body movement drops on the surface of TENGs. This corrosion may also be detrimental to the electronic output performance. Thus, a corrosion-resistant metal material is essential for wearable TENGs. Li et al. have recently proposed a humidity-resisting triboelectric nanogenerator that can provide stable output performance in high humidity environments. However, it is not based on a metal triboelectric material, and the output power density is relatively low, which greatly limits its application. However, roughening triboelectric surfaces can be used as a practical method to increase the electrical output of TENG. Therefore, it is urgent to propose a cost-effective, corrosion-resistant, conductor-to-dielectric TENG with high output performance. 

In this paper, we firstly developed a novel copper–nickel-based TENG (CN-TENG) composed of conductive copper–nickel alloy tape and polytetrafluoroethylene (PTFE) tape. Polymethyl methacrylate (PMMA) was utilized as the supporting structure. The copper–nickel alloy tape plays the role of the triboelectric pair and as conductive electrodes, due to its strong ability to lose electrons and its excellent electrical conductivity. Simultaneously, the rough surface of the copper–nickel alloy enables high output performance without additional surface roughening treatment, which significantly reduces processing costs and increases the output compared with other conductive metal, such as conductive copper, for example. Furthermore, the results obtained from a series of experiments revealed that the conductive copper–nickel alloy enabled the corrosion-resistant property of the proposed CN-TENG. The materials used for the TENG fabrication, including the alloy tape, PTFE tape, PMMA, and copper foil, are all common commodities in our daily life. The open-circuit voltage (V_OC_) and short-circuit current (I_SC_) can arrive at 196.8 V and 6 μA, respectively, and peak power density values of 45 μW/cm^2^ were realized for the CN-TENG. The experimental results revealed that the approximate value of V_OC_ of the fabricated CN-TENG integrated into the shoe reaches a maximum value of 1500 V, which is capable of driving at least 172 high-power light-emitting diodes (LEDs) in series.

## 2. Materials and Methods

The preparation process of CN-TENG is shown in Figure 1a. As demonstrated in Figure 1(a1), a piece of 3 cm × 6 cm transparent PMMA is used as a substrate. Subsequently, two pieces of 2 cm × 3 cm conductive copper–nickel alloy tape are attached on a PMMA substrate (see Figure 1(a2)). Then, the PTFE tape is applied to the surface of the copper foil, as illustrated in Figure 1(a3). Afterward, the substrate is folded into the device such that the PTFE tape surface faces the copper–nickel alloy tape surface, as illustrated in Figure 1(a4) and shown in Figure 1b. A stacked TENG is developed (see Figure 1(a5)) to strengthen the output current of the fabricated TENG. The SEM images in Figure 1c,d show the surfaces of the PTFE tape and the conductive tape. The fabric structure of the conductive copper–nickel alloy tape surface can increase the roughness of the triboelectric material, thereby enhancing the output performance of the TENG. The digital oscilloscope is utilized to measure electrical output. The fabricated TENG was activated by a vibrator (amplitude and frequency: 5 mm and 5 Hz, respectively), and a picture of the experimental setup is illustrated in Figure 1e.

## 3. Results and Discussion

Figure 2a shows a schematic illustrating the working mechanism of the CN-TENG. When PTFE film and conductive tape contact with each other, electrons will be injected from copper–nickel metal film into PTFE, in accordance with the electron-attraction ability of the materials. During full contact, the metal side is in electrical equilibrium. An electric potential difference can be constructed when PTFE separates from metal. This difference drives electron flow (owing to external loads) from the top electrode to the bottom electrode, and generates an electrical output signal. When the largest separation distance is reached in a complete contact–separation cycle, a new electrical equilibrium will be set up. Subsequently, if the PTFE is brought toward the conductive tape surface, then the electrons flow back to the top electrode, producing a reversed output current signal. The original state of the CN-TENG is restored with subsequent full contact between the PTFE and the conductive tape. Continuous electric output is engendered by periodic contact between the PTFE and the conductive tape. To elucidate the operation of the CN-TENG, the potential distribution was simulated using the COMSOL multi-physics software, as presented in Figure 2b. According to the simulation results, when displacement between two triboelectric materials increases, the potential difference also sharply increases.

The external force was provided by a vibration exciter to activate CN-TENG. As shown in Figure 3a,b, the performance of TENG (the size: 2 cm × 3 cm) was characterized by approximate values of the V_OC_ (under the external load of 500 MΩ) and I_SC_ (under the external load of 100 KΩ), which reached values of 196.8 V and 6 μA, respectively. As illustrated in Figure 3c, for resistance lower than 100 KΩ, the output current was ~6 μA. The output voltage increased to >196.8 V when the resistance increased from 100 KΩ to 500 MΩ. The maximum power density can arrive at 45 μW/cm^2^, which occurred at 30 MΩ, as shown in Figure 3d.

To improve the electrical output, a stacked CN-TENG was developed. The output performance of stacked TENGs with different numbers of working units were measured, and approximate I_SC_ values of 10.2 μA, 13.7 μA, and 18.2 μA, respectively, were obtained (see Figure 4a–c). Additionally, the output current obtained for different numbers of units was compared, as shown in Figure 4d. According to the results, the output current increased with as the number of units increased.

We compared the charging ability of CN-TENG and conventional TENG (based on conductive copper) by charging a 1-nF capacitor through a full-wave rectifier bridge. As illustrated in Figure 5a, at the peak value of the capacitor voltage (i.e., 21 V), 21 nC of charge was transferred for CN-TENG. However, for conventional TENG, only 13 nC of charge was transferred (see Figure 5b). The relatively high charge transfer (21 nC) is attributed to the rough surface of the conductive copper–nickel alloy tape, which produces more induced triboelectric charges due to the increased contact area. Moreover, the fabricated CN-TENG was tested by using a vibration platform, and the results revealed that the output voltage of the TENG stays stable even after 5000 cycles, as shown in Figure 5c.

Regarding wearable electronics, the corrosion resistance of the device has a significant influence on the electrical output. This investigation is important, as humans will sweat profusely during exercise activities. Therefore, the electrical output performance and the conventional copper TENG subjected to a corrosive environment was compared. In detail, both devices were immersed in a sodium chloride solution (mass fraction: 5%, 15%, and 25%) for 10 h, followed by subsequent exposure to air for 2 h. Afterward, the output performance was measured.

Figure 6 shows the electrical measurement of copper TENGs treated with different concentrations of sodium chloride solution. Approximate V_OC_ values of 150 V, 101 V, and 70 V; I_SC_ values of 5 μA, 3.4 μA, and 2.1 μA, and output power values of 187 μW, 93 μW, and 40 μW were realized at the aforementioned mass fractions (see Figure 6a–f). Corresponding values of 160 V, 5.7 μA, and 220 μW, which were obtained for the untreated TENG, were set as the reference values. Additionally, approximate V_OC_ values, I_SC_ values, and the maximum output power value of TENGs smeared with different mass fractions of the chloride solution were compared, as illustrated in Figure 6g–i. The results display that the output performance decreased with the increasing mass fraction of the solution. This decrease can be attributed to the weak corrosion resistance of the conductive copper.

The investigations under the same experimental conditions were performed on the CN-TENG. Based on the experimental results, approximate V_OC_ and I_SC_ values were measured, as shown in Figure 7a–f. Approximate V_OC_ values of 195 V, 191 V, and 187 V; I_SC_ values of 5.8 μA, 5.7 μA, and 5.5 μA; and output power values of 268 μW, 252 μW, and 241 μW, were realized for the treated CN-TENGs.

The aforementioned experimental results revealed approximate V_OC_, I_SC_, and maximum output power values of 196.8 V, 6 μA, and 270 μW, respectively, for the untreated CN-TENG, and these values were set as the reference values. Additionally, the approximate V_OC_, I_SC_, and maximum output power values of the TENGs smeared with different mass fractions of the chloride solution were compared, as presented in Figure 7g–i. According to the figure, each of these values decreased only slightly as the mass fraction of the solution increased. This is attributed to the excellent corrosion resistance of the conductive copper tape with nicked, compared with that of the conductive copper tape. It is noted that copper will undergo electrochemical corrosion in sodium chloride solution [41]. The related chemical reaction equations are as follows.

2Cu + H_2_O + CO_2_ + O_2_ = Cu_2_(OH)_2_CO_3_

The basic cupric carbonate produced by the reaction not only affects the electrical output of TENG, but also damages human health. However, the copper–nickel alloy has good corrosion resistance in the corresponding environment [42]. In order to visualize the corrosion on the surface of the copper foil, we analyzed the elements on the copper foil surface by sodium chloride solution (shown in Figure 8b), and compared the copper foil surface before treatment (shown in Figure 8a). According to the results, the proportion of copper elements on the surface of the treated copper decreased obviously, which will influence the electrical output of the TENG.

In addition, we demonstrated that the 5 cm × 20 cm CN-TENG can be easily integrated into a shoe, as shown in Figure 9b. When people are exercising, the CN-TENG integrated into shoe can be utilized to harvest mechanical energy, as shown in Figure 9a. The frequency of human motion is about 4 Hz, and the test time is 5 s. As the experimental results show (see Figure 9c), an approximate V_OC_ value of 1500 V is realized for the TENG in the shoe sole. Owing to the walking action, the TENG is capable of driving at least 172 high-power LEDs in series (working voltage: 3.4 V), as shown in Figure 9d,e.

## 4. Conclusions

A novel CN-TENG was developed using conductive copper–nickel alloy tape and PTFE tape. The copper–nickel alloy tape played the role of the triboelectric pair and conductive electrodes; PMMA was utilized as the supporting structure. The materials utilized for the fabricated TENG, including the alloy tape, PTFE tape, PMMA, and copper foil, are all common commodities in our daily life. The V_OC_ and I_SC_ values can arrive at 196.8 V and 6 μA, respectively, and peak power density values of 45 μW/cm^2^ were realized for the CN-TENG. To boost the output current, a stacked CN-TENG was fabricated, and the enhanced corresponding output performance was observed. A series of experiments demonstrate the excellent corrosion resistance of the proposed CN-TENG. In addition, we demonstrated that the TENG can be easily integrated into a shoe and operated by the movement of walking. From the experimental results, the approximate V_OC_ value of the fabricated TENG in the shoe reached 1500 V, which is capable of driving at least 172 high-power LEDs in series. The proposed TENG can be applied in the field of wearable electronics and serve as a continuous energy supply.

## Figures and Tables

**Figure 1 nanomaterials-09-00700-f001:**
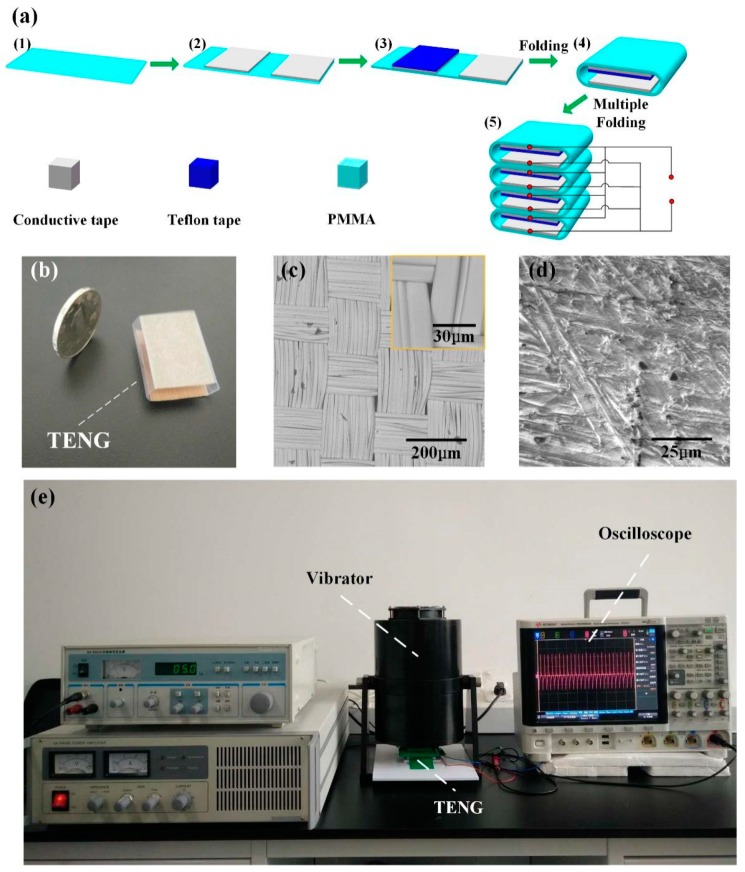
(**a**) Preparation process of the copper–nickel-based triboelectric nanogenerator (CN-TENG). (**b**) Photograph of the fabricated CN-TENG unit. SEM image showing the surface of the (**c**) conductive tape and (**d**) polytetrafluoroethylene (PTFE) tape. (**e**) The mechanical vibration system and measurement system.

**Figure 2 nanomaterials-09-00700-f002:**
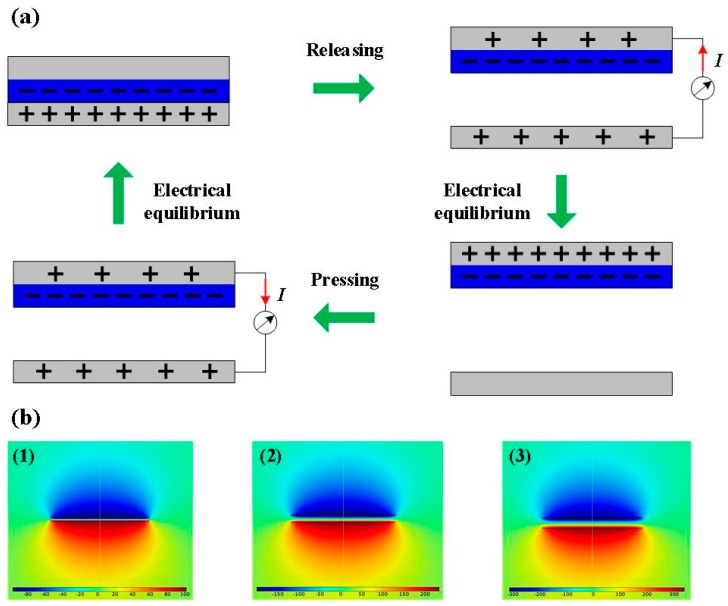
(**a**) Working mechanism of the CN-TENG. (**b**) Numerical calculations (performed in COMSOL) of the potential distribution across the electrodes of the TENG at each step (1–3) under open-circuit conditions.

**Figure 3 nanomaterials-09-00700-f003:**
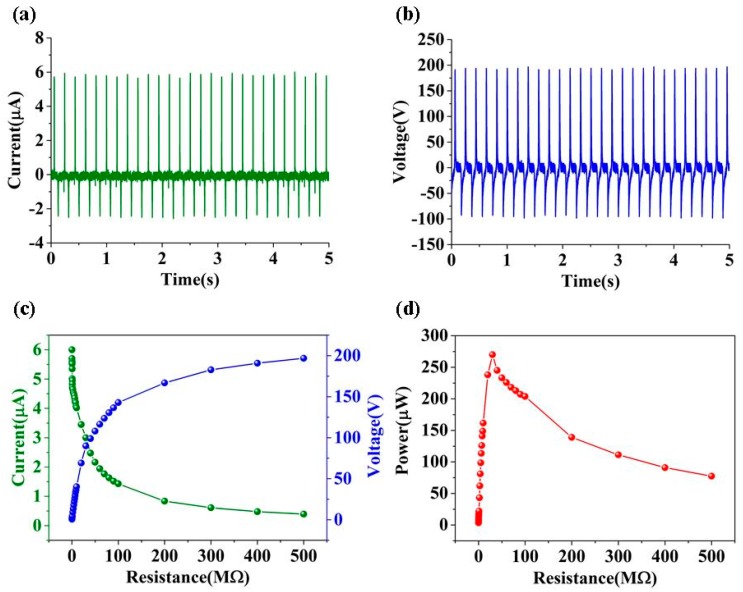
The (**a**) output current, (**b**) voltage of the TENG unit, (**c**) output voltage and current versus different load resistances, and (**d**) corresponding output power density under various load resistances.

**Figure 4 nanomaterials-09-00700-f004:**
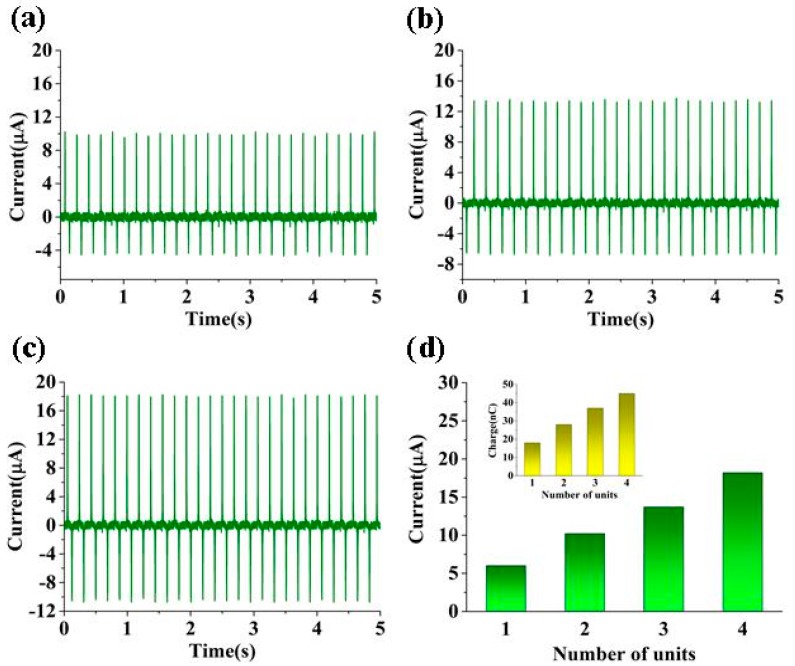
Characterization of a CN-TENG output with different unit numbers. The approximate I_SC_ value of the CN-TENG with (**a**) two units, (**b**) three units, and (**c**) four units. (**d**) Comparison of output performance.

**Figure 5 nanomaterials-09-00700-f005:**
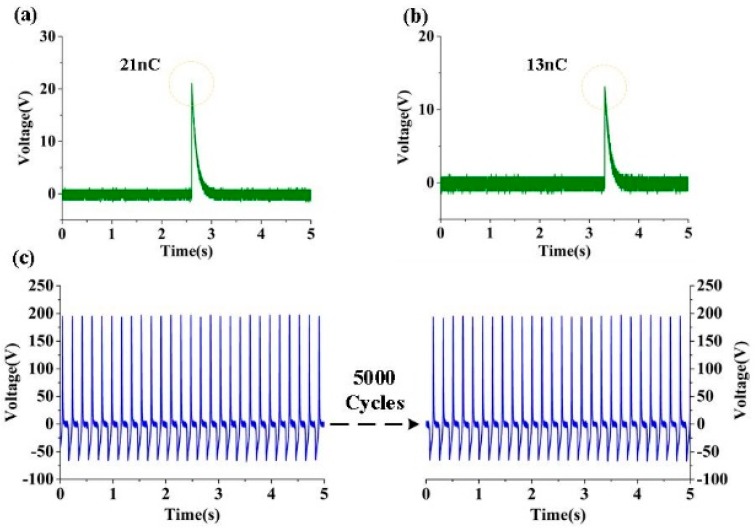
(**a**) The voltage curve of a 1-nF capacitor connected to the TENGs through a full-wave rectifier bridge, and (**b**) conductive copper tape, respectively, which serve as the triboelectric pairs. (**c**) The reliability of the prepared TENG was verified via 5000 cycles of continuous operation.

**Figure 6 nanomaterials-09-00700-f006:**
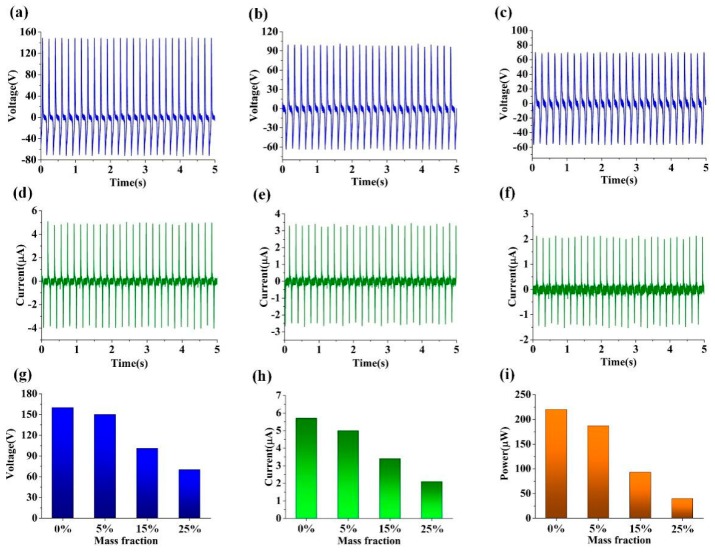
Electrical measurements of three copper TENGs treated with different concentrations of a sodium chloride solution. (**a**–**c**) Approximate open-circuit voltage (Voc) and (**d**–**f**) short-circuit current (I_SC_) values of the treated copper TENGs. Comparison of the approximate Voc (**g**), I_SC_ (**h**), and value of max output (**i**) power for the copper TENGs.

**Figure 7 nanomaterials-09-00700-f007:**
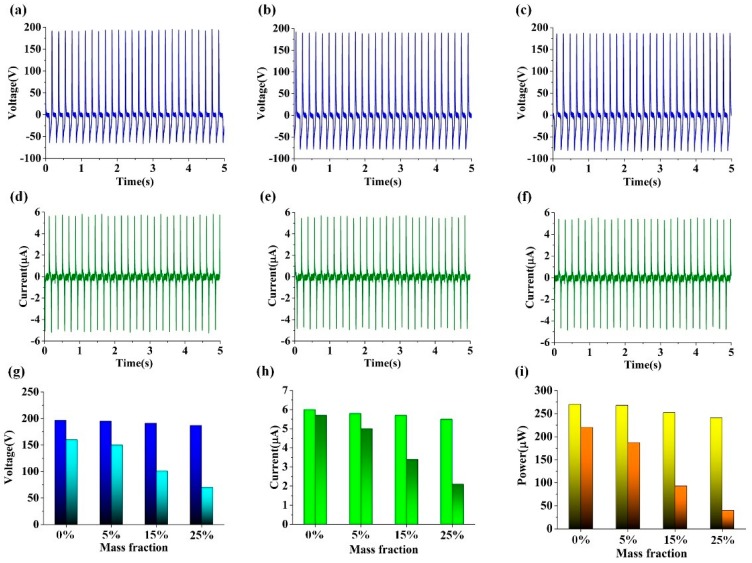
Electrical measurement of three CN-TENGs treated with different concentrations of sodium chloride solution. (**a**–**c**) Approximate V_OC_ and (**d**–**f**) I_SC_ values of the treated CN-TENGs. Comparison of the approximate V_OC_ (**g**), I_SC_ (**h**), and the value of max output (**i**) power for the CN-TENGs.

**Figure 8 nanomaterials-09-00700-f008:**
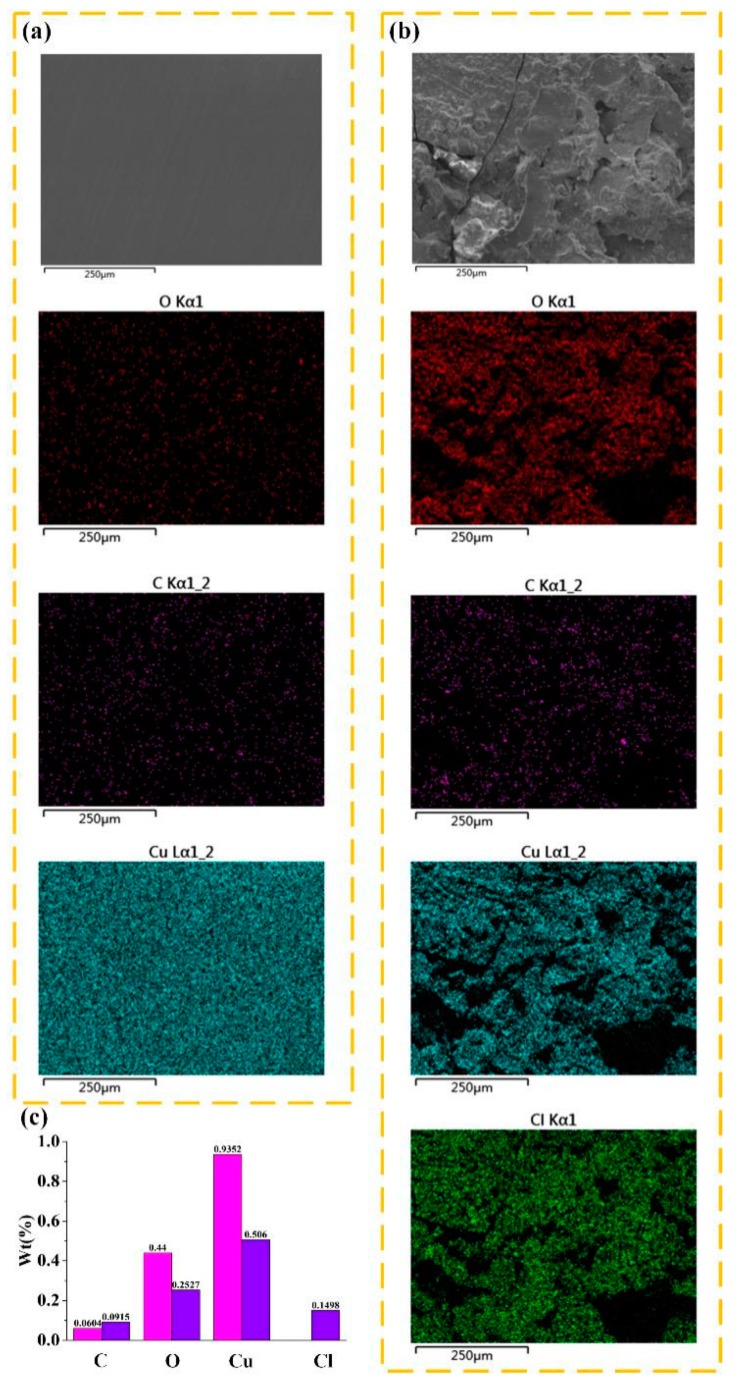
Elemental analysis of (**a**) the surface of the copper foil and (**b**) the copper foil surface treated with sodium chloride solution. (**c**) Comparison of element proportion on the copper foil surface treated with sodium chloride solution.

**Figure 9 nanomaterials-09-00700-f009:**
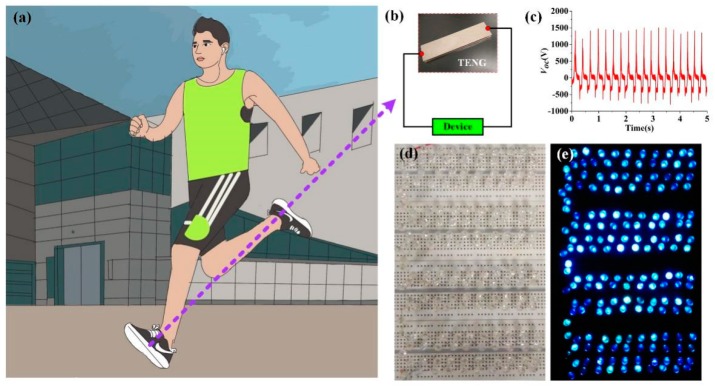
(**a**) The schematic of human walking. (**b**) Picture of the 5 cm × 20 cm fabricated TENG integrated into a shoe. (**c**) Approximate VOC of the TENG generated during human walking. (**d**,**e**) Photograph of the 172 high-power light-emitting diodes (LEDs) easily driven by the TENG during human walking.

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
