# Peer review of "Cost-Effective Copper–Nickel-Based Triboelectric Nanogenerator for Corrosion-Resistant and High-Output Self-Powered Wearable Electronic Systems"

_nanomaterials, 2019, doi:10.3390/nano9050700_

Round 1
Reviewer 1 Report
The authors have presented a corrosion-resistant copper-nickel based TENG (CN-TENG) intended for human motion application. The CN-TENG was analyzed, fabricated and tested on both benchtop and on human subject. This is a very interesting work and has technical merit. However, I have some concern about the content and presentation which must be addressed before acceptance. Please see my comments below:
1. Page 1, Abstract: “The approximate value of VOC for the fabricated TENG integrated into 23 the shoe reached 1500 V, which is capable of driving 172 high-power LEDs.”
- This statement misleads. What was the max. number of LEDs the authors tested? It might turn 500/1000/1720 LEDs depending how are they connected (series/parallel). Therefore, I would suggest putting the term ‘at least’ before ‘172 high-power LEDs’. Also, please add how those LEDs were connected (series/parallel)
2. Page 1, Introduction: “Vibration energy harvesters based on piezoelectric and electromagnetic effects are considered promising power supply sources for micro-devices.”
- Please justify this statement.
3. Page 1, Introduction: “…………value of 1500 V, which is capable of driving 172 high-power LEDs.”
- Please update based on comment no. 1.
4. Page 2, Materials and Methods: “The fabricated TENG was activated by a vibrator (amplitude and frequency: 5 mm and 5 Hz, 93 respectively).”
- Why did the authors choose this vibration frequency and amplitude. Any particular reason?
5. Page 4, Results and discussion: Fig. 2 and related discussion shows the FEA simulation. This is a very nice image; however, it is expected to have more discussion on the potential distribution, particularly in terms of displacement.
6. Please add the figure(s) to show the experimental setup (block diagram and or photograph).
7. Page 3-4, Results and discussion: “………of the V_OC (under the external load of 50 MΩ) and I_SC (under the external load of 100 kΩ), which……..”
- Why did the authors choose 50 M-ohm and 100k-ohm values. V_OC should be measured with high impedance probe (at least 1 G-ohm). And, I_SC should also be measured with its optimum load which seems to be 30 or 50 M-ohm as seen in Fig. 3(d). Rest of the lines in this page are not clear, must be updated with clear information (e.g., Fig 3 (d) indicates power, not power density)
8. Page 5, Results and discussion: “To improve the electrical output, a stacked CN-TENG was developed.””
- Please make it clear how was the electrical connection made out of the stack to achieve higher current.
9. Page 11: Please add more information about the human motion test setup including schematic connection diagram, walking/running speed, time etc.
10. Require English correction throughout the manuscript.
Author Response
We have revised it according to the reviewer's opinion. The specific information is in the document.

Reviewer 2 Report
Authors brings solution to corrosion problem in low-cost triboelectric devices using Cu-Ni electrodes. They use their TENG with high performance (1500 V) for wearable applications where sweat and other compounds can cause corrosion on electrodes. They don’t use any post-process for surface structuring which reduces fabrication time and cost. They use COMSOL to show the working principle of the TENG with success. Results are accurate and the article was writing scientifically correct. Experiments were designed correctly and enough detail was presented for repeatability. However, I have several concerns as described below:
1) Comparing Cu-Ni and Cu using a corrosion cell experiment (cyclic C-V curve) is very important for readers to see the difference.
2) Multilayerd TENG usually show multiple peaks Kanik et al, A Motion‐ and Sound‐Activated, 3D‐Printed, Chalcogenide‐Based Triboelectric Nanogenerator, Advanced Materials, 2015. Why they don’t observe the same behavior compare to the work by Kanik et al.
Author Response

(The authors gave the same response as above.)

Round 2
Reviewer 1 Report
The authors have addressed all my comments and revised the manuscript accordingly. I recommend accept as it is.